# Clinical Equivalence between Generic Versus Branded Antibiotics: Systematic Review and Meta-Analysis

**DOI:** 10.3390/antibiotics12050935

**Published:** 2023-05-21

**Authors:** André Cotia, Haliton Alves Oliveira Junior, Jessica Y. Matuoka, Ícaro Boszczowski

**Affiliations:** 1MBA Program in Prevention of Healthcare Acquired Infections, Infectious Diseases Department, Faculdade de Medicina da Universidade de São Paulo, São Paulo 05403-000, Brazil; 2Health Technology Assessment Unit, Hospital Alemão Oswaldo Cruz, São Paulo 01323-903, Brazil; haoliveira@haoc.com.br (H.A.O.J.); jessica_matuoka@hotmail.com (J.Y.M.); 3Infection Control Department, Hospital Alemão Oswaldo Cruz, São Paulo 01323-020, Brazil; icaro.b@hc.fm.usp.br; 4Infection Control Department, Central Institute, Clinics Hospital, Medicine Faculty, Universidade de São Paulo, São Paulo 05403-010, Brazil

**Keywords:** anti-bacterial agents, generic antibiotics, clinical equivalence, efficacy, systematic review

## Abstract

Regulatory authorities authorize the clinical use of generic drugs (GD) based on bioequivalence studies, which consist of the evaluation of pharmacokinetics after a single dose in vitro or in healthy individuals. There are few data on clinical equivalence between generic and branded antibiotics. Our aim was to synthesize and analyze the available evidence on the clinical efficacy and safety of generic antibiotics compared to their original formulations. A systematic review was performed on *Medline* (*PubMed*) and *Embase* and validated through *Epistemonikos* and *Google Scholar*. The last search was conducted on 30 June 2022. Meta-analyses of clinical cure and mortality outcomes were performed. One randomized clinical trial (RCT) and 10 non-randomized intervention studies were included. No differences in clinical cure were observed between groups in the meta-analysis (OR = 0.89, 95% CI [0.61–1.28]; I^2^ = 70%, *p* = 0.005). No difference was observed between groups when considering the use of carbapenems for overall mortality (OR = 0.99, 95% CI [0.63–1.55]; I^2^ = 78%) or death associated with infections (OR = 0.79, 95% CI [0.48–1.29], I^2^ = 67%). Most of the studies were observational, and the duration of follow-up, the characteristics of the participants, and the sites of infections were heterogeneous. Due to the uncertainty of the evidence, it is not possible to contraindicate the use of generics, which is an important strategy to expand access.

## 1. Introduction

Access to antibiotic drugs is a global challenge. The World Health Organization (WHO) estimates that one-third of the population or up to one-half in some geographic areas does not have access to antibiotics and other essential medicines [1]. For this reason, it is crucial to establish public policies to solve this problem. Some countries have overcome this issue by making antibiotics available through a list of essential medicines [2]. Nevertheless, the consistency of these policies is threatened by increasing costs. Even in areas where these policies are in place, it is not uncommon for patients to be unable to access antibiotics due to their limited availability in government programs [1].

The bioequivalent generic antibiotics are often less expensive than the branded drugs, as the development and research costs are borne by the company of the branded formulation [3]. Moreover, in Brazil, a current policy limits the marketing price of generics to 35% of their branded counterparts [4]. Thus, a very well-established regulation for the production and marketing of generic antibiotics contributes to guaranteeing wider access to these drugs in concert with all investments made by the pharmaceutical researchers and developers [5].

Governmental and/or independent regulatory agencies such as the Food and Drug Administration (FDA), the European Medicines Agency (EMA), the World Health Organization (WHO), as well as other similar institutions in different countries have established rules that certify a given generic medicine, including antibiotics, as bioequivalent to its respective brand drug [6,7]. The regulatory rules between countries are quite similar, and minor differences are described in their dedicated documents and websites [5,6]. Most of the bioequivalence studies are in vitro and based on pharmacodynamic parameters after a single dose administered to healthy individuals [8]. Briefly, two products are bioequivalent if the 90% confidence interval for the mean difference between the area under the curve and the maximum serum concentration for both do not exceed ± 20% (80% to 120%) or 80% to 125% if the parameters are logarithmically transformed [2]. This method is largely accepted by most of the regulatory agencies to compare products with the same active ingredient and to authorize them for clinical use [8].

The sale of generics accounts for 60% of all medicines in the United States of America and 35% in Brazil [9,10]. In the latter country, the price of antibiotics for the consumer decreased up to 62% after the implementation of a law that regulates the sales of generics [11]. Likewise, there are estimates of savings of USD 1.3 billion in healthcare expenditures following the implementation of a policy for generic medicines in France [4].

Despite the economic advantages and advances in equity provided by generics availability, data about clinical equivalence are still scarce. In theory, small differences between their pharmaceutical or physicochemical properties (e.g., a salt or ester) could result in altered pharmacokinetic/pharmacodynamic parameters and compromise their efficacy/effectiveness [2,7,12,13].

Clinical studies, which are mandatory for brand-name drug approval, are not required for the approval of generics. It is assumed that the bioequivalence of the active ingredients is a surrogate of the therapeutic or clinical equivalence [14]. There are few studies based on clinical equivalence, i.e., trials to test non-inferiority between these formulations [2,8,15].

The aim of this study is to systematically review the literature and synthesize the available evidence about the clinical equivalence and safety of generic antibiotics compared to their branded counterparts.

## 2. Method

### 2.1. Study Design

We performed a systematic review of the literature including comparative studies between generic antibiotics and the correspondent branded drugs regarding their clinical equivalence. The research question that guided this systematic review was the following: What is the efficacy, safety, and clinical equivalence of generic antibiotics compared to the correspondent brand drugs? The core elements that guided our search were the following:P (population): Individuals who received antibiotics for prophylaxis and/or therapy for bacterial infections;I (intervention): Prescribing antibiotics for prophylaxis and/or treatment of infections based on syndromic or microbiological diagnosis;C (comparison): Generic antibiotics, defined as those drugs with proven bioequivalence to the branded drug;O (outcomes): The primary outcomes of interest are clinical cure, microbiological cure, and mortality. Secondary outcomes are adverse events switch-back rate, adherence, pharmacotherapy drop-out, hospital admission, and recurrent prescription of antibiotics.

Eligibility criteria. Due to the scarcity of studies found in a validation search, any study design was considered for inclusion as long as it provided a comparison between the generic and its respective branded formulation in a superiority or non-inferiority study. We excluded phase II randomized clinical trials, bioequivalence and pharmacokinetics assays, animal models, and in vitro studies. The year of publication was not an exclusion criterion. Only studies published in English, Portuguese, or Spanish were considered for inclusion.

Electronic search. The literature search was conducted on *Medline* (*PubMed*) (pubmed.ncbi.nlm.nih.gov) and *Embase* (embase.com) and validated at *Epistemonikos (epistemonikos.org)* and *Google Scholar (scholar.google.com)* to ensure that all relevant studies would be included. Furthermore, we manually searched the list of references of the selected studies. The date of the last search was 30 June 2022. We matched the terms related to generic and branded formulations using appropriate Boolean operators. The search strategy is provided in Appendix A.

### 2.2. Selection and Data Extraction

After excluding duplicates, two independent reviewers (A.L.F.C. and I.B.); selected all references by titles and abstracts using the webapp Rayyan QRCI^®^. Potentially eligible studies had their full texts evaluated for inclusion, and in case of conflict, a third reviewer (J.Y.M or H.A.O.J.) was consulted.

The two independent reviewers extracted the information in a spreadsheet (Microsoft Excel^®,^ Microsoft Corporation, Redmond, WA, USA). Extracted data included characteristics of the study (author and year of publication, setting and location of the study, length of follow up, and antibiotic type), characteristics of the participants, and results on efficacy and safety (clinical cure, microbiologic cure, mortality, and adverse events). To ensure the quality of abstracted data, a third reviewer validated the process of data abstraction.

### 2.3. Risk of Bias

Bias was evaluated by two independent reviewers (H.A.O.J. and J.Y.M.) using the following standardized tools recommended by the Cochrane Collaboration: the Risk of Bias version 2 (RoB-2) for randomized clinical trials and the Risk of Bias in Non-Randomized Studies of Intervention (ROBINS-I) for other comparative non-randomized studies of intervention.

### 2.4. Synthesis of Results

In this study, clinical equivalence was defined by the similarity between treatments in comparative studies for human treatment or prophylaxis when considering clinical meaningful outcomes [16].

The primary outcomes evaluated were clinical cure, microbiological cure, and mortality.

Clinical cure is defined as the investigator’s assessment of clinical response. Microbiological cure comprises time to negativity of blood cultures and decrease in bacterial count of quantitative cultures of different biological samples. Mortality was evaluated as overall mortality or death associated with infections after a proper antibiotic regimen. Any adverse events are defined as the number of individuals with at least one adverse event. Serious adverse events are defined as the number of individuals with at least one serious adverse event.

Given the heterogeneity of the studies, the results were summarized in a pairwise meta-analysis of random effects. We calculated an estimate of the effects of all studies combined for those reporting clinical cure rates, and we performed an additional sensitivity analysis, stratifying it according to the study design and class of antibiotics. Heterogeneity was verified by the Higgins test I^2^.

We did not analyze the possibility of publication bias due to the small number of included studies in each meta-analysis (<10 studies) [17,18,19]. Furthermore, whenever significant heterogeneity is present, none of the proposed tests show uniform and good properties [18].

We did not perform the analysis of adverse events due to the inconsistent report in the studies included in this review.

### 2.5. Quality of the Evidence

The body of evidence for all relevant outcomes was assessed using criteria established by the Grading of Recommendations Assessment, Development, and Evaluation (GRADE) [20].

## 3. Results

### 3.1. Study Selection

The literature search retrieved 274 studies after excluding seven duplicates. Those 274 were screened by their titles and abstracts. Twenty-three titles had the full text assessed for inclusion. Twelve studies were excluded during this step: four due to study design, three because of the participants’ category, three due to definition of outcomes, one study with only the abstract in English and the full article written in Japanese, and one due to lack of full-text publication. At the end of the selection process, eleven studies were included (Figure 1).

### 3.2. Characteristics of the Studies

We present a summary of the studies in Table 1. Among the eleven included studies, only one was a single-blind randomized clinical trial, and ten were observational studies. Considering the observational studies, eight were retrospective, and two had a retrospective and prospective phase.

All studies were published in the last 15 years: six between 2008 and 2012, two between 2015 and 2016, and three after 2019. Four studies were carried out in Thailand and the others in Indonesia, Japan, Taiwan, Colombia, Chile, South Africa, and Greece.

Methods and outcomes varied, and the criteria for clinical cure or improvement were not described in all studies. Among the six studies that assessed clinical cure, generic antibiotics were not different from their respective branded formulations. Carbapenems were the most frequently investigated class of antibiotics: four studies compared meropenem formulations, and two compared imipenem formulations. The other studies compared formulations of cefuroxime, cefoperazone-sulbactam, ciprofloxacin, piperacillin-tazobactam, clarithromycin, trimethropim-sulfamethoxazole, pipemidic acid, and teicoplanin.

The clinical characteristics of the compared populations (generic versus branded) were similar except for Angkasekwinai et al. (2011) [21], which had a statistically significant difference between the groups in relation to more patients with heart disease in the group that received original meropenem and more patients using immunosuppressive drugs in the group that received generic meropenem.

The studies had a heterogeneous population. Araya et al. (2015) [22] analyzed cancer patients, and Punpanich et al. (2012) [23] was the only that included patients under 18 years. Snyman et al. (2009) [6] and Lin et al. (2016) [24] included outpatients, while the others only included inpatients.

Ten studies analyzed antibiotics prescribed for treatment, and one, namely Mastoraki et al. (2008) [25], analyzed surgical prophylaxis. Among those that analyzed treatment, the site of infection varied. Snyman et al. (2009) [6] assessed cases of upper respiratory infection and pneumonia with clarithromycin. Handayany et al. (2020) [26] studied typhoid fever; Lin et al. (2016) [24] included outpatients with a diagnosis of urinary tract infections. The other studies included patients with diagnosed infections in different sites, for example, respiratory and urinary tracts, bloodstream, intra-abdominal, central nervous system, skin and soft tissue infections, and febrile neutropenia.

### 3.3. Risk of Bias

The only RCT had high risk of bias mainly due to missing outcome data and possible selection on the reported results. In addition, a generic manufacturing company sponsored the study, and the authors did not declare the level of their participation in the study.

The observational studies had serious or critical risk of bias mainly in three domains: 80% of bias due to confounding and 70% of bias due to missing data and bias in measurement of outcomes. For the other three domains, risks were low and moderate: 100% of bias in classification of interventions, 80% of bias in selection of the reported results, and 70% of bias due to deviations from intended interventions. Regarding bias due to selection of participants, 50% of the studies had low or moderate risk, and the other half had serious or critical risk of bias.

Detailed information on risk of bias is available in Figure 2.

### 3.4. Summary of Results

Seven studies showed no statistically significant difference in primary outcomes comparing generic antibiotics and their respective branded counterparts [6,21,22,23,24,27,28].

Three studies [24,26,29] did not report microbiological data, and none were used in the meta-analysis. The other studies reported microbiological data; however, in the study by Araya et al. (2015) [22], antimicrobial resistance data were not reported.

Angkasekwinai et al. (2011) [21] isolated microorganisms from two-thirds of the patients. The most common pathogen was Gram-negative bacteria, and susceptibility to meropenem of these isolates was available in 96% percent. No significant difference in susceptibility to meropenem was found between original and generic group except for *A. baumannii*; the isolates in the branded group were more susceptible to meropenem than those in the generic group (30% vs. 8.1%, *p* = 0.02).

Piyasirilp et al. (2010) [27] observed documented microbiological infections in two-thirds of the patients. All isolates of extended-spectrum-beta-lactamase (ESBL) producing *E. coli* and *K. pneumoniae* were susceptible to imipenem/cilastatin. *P. aeruginosa* isolates were susceptible to imipenem in 84.8% and 82.9% of the patients in the original and the generic groups, respectively. *A. baumannii* isolates were susceptible to imipenem in 65.5% and 60% of the patients in the original and the generic groups, respectively. The characteristics of infections and causative agents of infections of the patients in both groups were not significantly different.

Snyman et al. (2009) [6] collected approximately 164 respiratory samples (110 throat swabs and 54 sputum specimens) from patients with tonsillitis and pneumonia. The organisms cultured included *S. pyogenes*, *H. influenzae*, *S. aureus*, *S. pneumoniae*, and *H. parainfluenzae*. The results of the bacteriological cure rates showed similarly high rates.

Tansuphasawaikul et al. (2011) [28] showed no statistical differences of the distribution of the sites of infection and causative microorganisms between the two groups except for ESBL-producing *E. coli*, which was significantly higher in the generic meropenem group.

Ordonez et al. (2019) [30] included only patients with documented Gram-negative infections susceptible to meropenem to maximize the likelihood of successful treatment with the selected antibiotics.

Punpanich et al. (2012) [23] observed microbiologically confirmed infections in 40% and 32% among the generic and original meropenem groups, respectively. Including all sites of positive culture, the rate identification (infection/colonization) of ESBL-producing Gram-negative enteric bacteria and multi-drug resistant *Acinetobacter baumannii* was not significantly different between the two treatment groups. Microbiological cure rates were also comparable for the generic and original groups, respectively.

Clinical cure was reported in six studies that included 2372 patients. There was no significant difference between generic and branded antibiotics (OR = 0.89, IC 95%, [0.61–1.28]). However, this subgroup showed significant heterogeneity (I^2^ = 70%, *p* = 0.005) (Figure 3).

We performed a sensitivity analysis to investigate the influence of study design (RCT or observational) on the results, but no changes were observed. The test for subgroup differences was not statistically significant (Figure 4).

Further analyses were performed comparing generic and branded carbapenems in retrospective observational studies. There was no difference for overall mortality and infection-associated death after treatment (Figure 5 and Figure 6). However, there was high heterogeneity in the analyses.

### 3.5. Quality of Evidence

The quality of evidence from the randomized clinical trial was low. It was downgraded by two levels due to serious risk of bias and imprecision. Regarding the analysis of observational studies for clinical cure and all-cause and infection-related mortality, the quality of evidence was very low. For this set of studies, the certainty of evidence was downgraded by three levels due to very serious risk of bias, inconsistency, and imprecision. The population included in the studies was highly heterogeneous (clinically and statistically), and the total number of patients included in each outcome analysis was not enough to precisely evaluate the effect size. The complete analysis and explanations are presented in Table 1.

**Table 1 antibiotics-12-00935-t001:** Summary of clinical equivalence studies comparing generic and branded antimicrobials (2008–2020). Punpanich 2012 [23], Ordoñez 2019 [30], Lin 2016 [24], Araya 2015 [22], Mastoraki 2008 [25], Piyasirisilp 2010 [27], Angkasekwinai 2011 [21], Tansuphasawadikul 2011 [28], Shinoda 2019 [29], Hadayany 2020 [26], Snyman 2009 [6].

Study		*n*	Male (%)	Age, Mean ± SD	Compared Drugs	Type of Infection	Design	Main Findings	Definitions
Mastoraki et al. (2008)	Generic Branded	305 313	259 (84.9%) 266 (85%)	63.8 ± 7.7 64.3 ± 7.1	Intravenous cefuroxime (Normafenac^®^), Intravenous cefuroxime (Zinacef^®^) Both received a 3 g single dose as surgical prophylaxis immediately before anesthetic induction. Complicated cases (as stated by authors) received a supplemental doses up to 12 h after the surgery.	Antibiotic prophylaxis before myocardial revascularization.	Quasi-experimental study (before–after). Patients received generic cefuroxime for 4 weeks and then branded cefuroxime for other another 4 weeks. Data were obtained retrospectively.	Surgical site infection rates, generic vs. branded (*n* = 31, 10.1% vs. *n* = 6, 1.9%, respectively) *p* < 0.0001. Total postoperative infections, generic vs. branded (*n* = 39, 12.8% vs. *n* = 8, 2.5%, respectively) *p* < 0.0001. Death associated with infection—not reported. Crude mortality—not reported.	Surgical site infection, local erythema of the skin and soft tissues and/or pus associated with fever or hypothermia, white blood cell count > 10,000/mm^3^, and positive culture. Infections at other body sites or fluids were defined based on guidelines from the Centers for Disease Control and Prevention. Diagnosis of sepsis was based on the ACCP/SCCM criteria.
Snyman et al. (2009)	Generic Branded	141 136	64 (45%) 62 (46%)	34.68 ± 10.02 35.49 ± 10.56	Clarithromycin (Klarithran MR^®^), Ranbaxy Laboratories Ltd, Haryana, India Clarithromycin (Klacid XL^®^), Abbott Laboratories, Abbott Park, IL, USA 1 vs. 500 mg once a day, five days (tonsillitis); 1 vs. 500 mg once a day, for ten days (sinusitis and pneumonia).	Upper and lower respiratory infections.	Prospective randomize, comparative, investigator-blinded, and multicentric.	Clinical cure rate, (ITT), generic vs. branded (*n* = 114, 94.2% vs. *n* = 117, 95.9%, respectively) *p* = 0.57. Microbiological cure rate, generic vs. branded (*n* = 31, 79.5% vs. *n* = 39, 84.8%, respectively) *p* = 0.577. No deaths occurred during the study.	Clinical cure was defined as resolution of symptoms at the end of therapy. Respiratory specimens were collected at diagnosis and again at the follow-up visit to establish bacteriological cure, which was defined as eradication of the respiratory pathogen after the completion of antibacterial treatment. Sinusitis was diagnosed clinically and not confirmed with microbiological testing.
Piyasirisilp et al. (2010)	Generic Branded	300 300	154 (51.3%) 150 (50%) *p* = 0.81	66.5 ± 18 66.1 ± 17.8 *p* = 0.93	Imipenem/cilastatin (Yungin^®^) Imipenem/cilastatin (Tienam^®^)	Different sites, including community acquired-infections and healthcare-acquired infections.	Observational, retrospective, descriptive. There were two cohorts with patients who received antimicrobials for treatment > 48 h with generic or branded in each.	Clinical cure rate, generic vs. branded (*n* = 86, 28.7% vs. *n* = 105, 35%, respectively) *p* = 0.09. Microbiological cure rate, generic vs. branded (*n* = 133, 44.3% vs. *n* = 147, 49%, respectively) *p* = 0.29. Death associated with infection, generic vs. branded (*n* = 64, 21.3% vs. *n* = 55, 18.3%, respectively) *p* = 0.41. Crude mortality generic vs. branded (*n* = 129, 43% vs. *n* = 106, 35.3%, respectively) *p* = 0.06.	Clinical cure, microbiological cure, and death associated with infection were not defined. Day at death associated with infections and crude mortality were not defined.
Tansuphasawadikul et al. (2011)	Generic Branded	207 190	124 (59.9%) 104 (54.7%) *p* = 0.30	65.7 ± 16.9 67.2 ± 17 *p* = 0.39	Mropenem (Mapenem^®^) Meropenem (Meronem^®^)	Different infection sites.	Retrospective cohort including inpatients. Multicentric.	Clinical cure rate at day 14, generic vs. branded (*n* = 64, 30.9% vs. *n* = 44, 23.2%, respectively) *p* = 0.08. Microbiological cure rate—not reported. Death associated with infection, generic vs. branded (*n* = 12, 5.8% vs. *n* = 21, 11.1%, respectively) *p*–not reported. Crude mortality, generic vs. branded (*n* = 22, 10.6% vs. *n* = 34, 17.9%, respectively) *p*–not reported.	Clinical cure was complete resolution of infection at day 14. Death associated with infection was not defined. Death was associated with infection and crude mortality at day 14.
Angkasekwinai et al. (2011)	Generic Branded	300 300	164 (54.7%) 145 (48.3%) *p* = 0.12	61.7 ± 19.8 64.6 ± 17.8 *p* = 0.06	Meropenem (Mapenem^®^) Meropenem (Meronem^®^)	Different sites, including community acquired-infections and healthcare-acquired infections.	Observational, retrospective, descriptive. There were two cohorts with patients who received antimicrobials for treatment > 48 h with generic or branded in each	Clinical cure rate, generic vs. branded (*n* = 211, 70.4% vs. *n* = 189, 63%, respectively) *p* = 0.07. Microbiological cure rate, generic vs. branded (*n* = 89, 29.7% vs. *n* = 69, 23%, respectively) *p* = 0.08. Death associated with infection, generic vs. branded (*n* =48, 16% vs. *n* = 65, 21,7%, respectively) *p* = 0.09. Crude mortality, generic vs. branded (*n* = 97, 32.3% vs. *n* = 114, 38%, respectively) *p* = 0.17.	Clinical cure was defined as cure and improvement at the end of treatment. Microbiological cure was not defined but described as eradicated. Death associated with infection was not defined. Day at death associated with infections and crude mortality were not defined.
Punpanich et al. (2012)	Generic Branded	180 180	95 (52.8%) 100 (55.6%) *p* = 0.59	6.72 (42) 2.95 (22.3) *p* = 0.016 Age in months	Meropenem (Mapenem^®^) Meropenem (Meronem^®^)	Different infection sites.	Observational, retrospective, and descriptive, including inpatients from 0 to 18 years of age.	Clinical cure rate at day 14, generic vs. branded (*n* = 18, 20% vs. *n* = 33, 32%, respectively) *p* = 0.29. Microbiological cure rate at day 7, generic vs. branded (*n* = 17, 80.9% vs. *n* = 13, 81.3%, respectively) *p* = 0.98. Proportions of those with positive initial culture only. Death associated with infection—not reported. Crude mortality at day 28, generic vs. branded (10% vs. 13.9%, respectively) *p* = 0.152—no reported absolute numbers.	Clinical cure was complete resolution of infection at day 14. Only pathogens isolated in normally sterile sites or those with higher than 10^5^ colony-forming units isolated in urine were regarded as true pathogens in the present study. Microbiological cure at day 7 of treatment.
Araya et al. (2015)	Generic Branded	135 206	51% 47% *p* = 0.53	59 ± 15.6 60 ± 12.6 *p* = 0.37	Cefoperazone-sulbactam (Cefactam^®^), Imipenem-cilastatine (Imipen^®^), and Piperaciline-tazobactam (Piperazam^®^), Libra Laboratories Cefoperazone-sulbactam (Sulperazone^®^), Pfizer Laboratories Imipenem-cilastatine (Tienam^®^), Merck Sharp & Dohme Corporation and Piperaciline-tazobactam (Tazonam^®^), Wyeth Laboratories	Different infection sites.	Comparative case series.	Clinical cure rate, all antibiotics, generic vs. branded (*n* = 105, 79% vs. *n* = 143, 69.5%, respectively) *p* = 0.10. Microbiological cure rate–not reported. Death associated with infection–not reported Crude mortality–not reported	Clinical cure not defined.
Lin et al. (2017)	Generic Branded	6900 2294	986 (14.3%) 492 (21.5%) *p* < 0.0001	Age (year) group: *n* 20–44: 3155 (45.7%) 45–64: 2549 (36.9%) ≥ 65: 1196 (17.3%) 20–44: 936 (40.8%) 45–6: 867 (37.8%) ≥ 65:91 (21.4%) *p* < 0.0001	Trimethoprim-sulfamethoxazole and pipemidic acid	Uncomplicated urinary tract infection.	Population-based retrospective cohort. “Failure” was assumed for those patients who were discharged from the emergency department (ER) with antimicrobials (generic or branded) for the treatment of uncomplicated urinary tract infections and required an additional outpatient visit for continuation of antimicrobials.	Treatment failure: ER or hospitalization due to UTI and required antibiotics generic vs. branded (*n* = 142, 2.1% vs. *n* = 78, 3.4%, respectively) *p =* 0.85. Further outpatient visit due to UTI and required antibiotics generic vs. branded (*n* = 1096, 16.2% vs. *n* = 351, 15.8%, respectively) *p* = 0.5. Microbiological cure rate—not reported. Death associated with infection—not reported. Crude mortality—not reported.	Treatment failure was determined in two ways: (i) an ER visit or hospitalization due to a UTI with antibiotic prescription within 42 days of the index consultation and (ii) an additional outpatient visit for a UTI requiring antibiotic treatment within 42 days of the completion of the original antibiotic therapy. Those who met the criteria of treatment failure (i) would not be enrolled for analysis of treatment failure (ii).
Shinoda et al. (2019)	Generic Branded	44 44	24 (55%) 25 (57%)	70.4 ± 13.2 74.4 ± 9.1 *p* = 0.19	Teicoplanin, Nanopia TDM Teicoplanin (Sekisui Medical Co. Ltd.) Teicoplanin, Targocid (Sanofi Co. Ltd.)	Not reported.	Observational, retrospective.	Initial mean serum concentration, generic vs. branded (12.8 mg/L vs. 16.3 mg/L, respectively) *p* < 0.01. Discontinuation rates due to adverse events, generic vs. branded (*n* = 1, 2.3% vs. *n* = 3, 4.8%, respectively) *p* = 0.64. Microbiological cure rate—not reported Death associated with infection—not reported. Crude mortality—not reported.	Discontinuation rate due to adverse events was determined by chart review.
Ordonez et al. (2019)	Generic Branded	68 100	42 (62%) 54 (54%)	58.6 ± 19.7 56.47 ± 17.45	Meropenem Meropenem (Meronem^®^)	Different infection sites.	Observational, retrospective, descriptive. Two cohorts of intensive care unit patients with meropenem susceptible Gram-negative bacilli infections.	Clinical cure rate—not reported. Microbiological cure rate—not reported. Death associated with infection—not reported. Crude mortality, generic vs. branded (*n* = 42, 62% vs. *n* = 41, 41%, respectively) *p* = 0.008.	Crude mortality at day 28.
Haydayany et al. (2020)	Generic Branded	34 33	Total: 37.3% masculine	Not reported	Ciprofloxacin Ciprofloxacin (Tequinol^®^)	Typhoid fever,	Observational, retrospective, descriptive. This study included patients from 15 to 60 years of age with a documented diagnosis of typhoid fever who received generic or branded antimicrobials.	The duration of fever reduction for patients using the Tequinol^®^ brand ranged from 1 to 3 days and for patients using ciprofloxacin generic ranged 3–7 days. Microbiological cure rate—not reported, Death associated with infection—not reported. Crude mortality—not reported.	Clinical cure defined as duration of fever reduction.

## 4. Discussion

The therapeutic equivalence between generics and innovative antibiotics has been questioned based on studies in animal models [2]. However, comparative studies analyzing generic and branded antibiotics are rare in the literature. We systematically reviewed the current literature and found that the data available to date are limited, heterogeneous, and at high risk of bias. Based on these data, it is not possible to state that generics are non-equivalent to their innovator for therapy or prophylaxis. For the outcomes considered, the results were imprecise, as not enough participants were included in the analyses. For these reasons, the certainty of the evidence varied between low to very low depending on the outcome and study design, meaning that the estimate of effect is uncertain, and future work may impact the overall conclusion.

This review included ten cohorts and one randomized clinical trial. There was high heterogeneity between the studies, as they included different populations, scenarios, drugs, sites of infection, duration of follow-up, and mainly different criteria for clinical cure. Seven of the eleven studies were on β-lactam antibiotics. Other classes of antibiotics analyzed in the included studies were glycopeptides, fluoroquinolones, trimethoprim-sulfamethoxazole, and macrolides.

Most studies had a high risk of bias. Observational studies compared outcomes in cohorts with historical controls, more than one class of antibiotics was prescribed concomitantly, and there were clinical differences between the groups or a lack of this information as well as a gap in the definition of outcomes. In addition, the branded antibiotic was compared with only one generic formulation, although there are often multiple generic antibiotics marketed for the same compound.

Interestingly, among the eleven studies included, only three compared oral antibiotics: Snyman et al. [6], Lin et al. [24], and Haydayany et al. [26] This aspect is of particular importance, as oral antibiotics may exhibit greater variability in absorption, dissolution, and other pharmacokinetic parameters compared to intravenous formulations [6,24,26].

A previous systematic review was published by Tattevin et al., in which in vivo studies (neutropenic mouse thigh infection model) demonstrated therapeutic inequivalence and lower bactericidal activity of generic antibiotics compared to their original formulation [2,31]. In this review, one study evaluated nine generic oxacillin products against the reference, and four were not therapeutic equivalents in microbiological assays. Subsequently, in the evaluation of treatment for *Staphylococcus aureus* infections using the neutropenic mouse thigh infection model, all generics did not reach Emax (maximum effect at log10 cfu/g): the product with the highest and lowest efficacy killed 22 and 1585 times less microorganisms per gram of tissue, respectively, than the reference drug [2,32]. Other studies evaluating the in vivo efficacy using the aforementioned model are cited in this systematic review. It is important to note that these models have not been validated for the evaluation of the efficacy of antibacterial drugs [2]. One study evaluated 46 generic piperacillin-tazobactam products from 17 countries, and incremental MIC antibiotic trials were performed, which were on average 10% lower than the innovator [33].

Most regulatory agencies require bioequivalence studies for approval of a generic antibiotic and waive clinical equivalence analysis. The production of generic antibiotics is an important strategy for expanding access, allowing affordable prices for larger populations. and promoting equity. Likewise, current regulatory requirements are important tools that ensure the quality of delivered generic drugs. Clinical trials comparing both formulations are difficult to be performed because of the competing interests of companies and the consumption time and cost. On the other hand, such studies would add evidence to answer the question of the real need to review the authorization process by regulatory agencies. Nonetheless, there is room for improvement in monitoring the safety and quality of these products in pharmaceutical and bioequivalence terms. In addition, pharmacovigilance in post-marketing medicines, generic or branded, is an important tool for continuous monitoring of quality and safety.

Our review has limitations. We compared a small number of studies, most of which were retrospective observational cohorts. The fact that we did not find a significant difference in the outcomes evaluated in this review should be interpreted with caution due to limitations in the design of the included studies. As previously pointed out by Tattevin et al. [2], like other systematic reviews, this work is subject to publication bias.

Finally, we believe that ensuring the production of generics is a way to guarantee the expansion of access to antibiotics as essential medicines to a very large number of people around the world, especially in low- and middle-income countries. In this sense, efforts to add new evidence of therapeutic equivalence between these formulations before a revision of authorization process is welcome and probably will need collaborative work among governments, pharmaceutical companies, and academic research centers.

## 5. Conclusions

This systematic review with meta-analysis showed no difference between generic and branded antibiotics in clinical cure or mortality. Most studies were observational and had a high risk of bias and heterogeneity. Well-designed and robust studies that address the issue of clinical equivalence between generic and branded antibiotics are needed.

## 6. Registration and Protocol

This systematic review of the literature was registered on the *PROSPERO* database (130893) and reported following the Preferred Reporting Items for Systematic Reviews and Meta-Analyses (PRISMA) Statement.

## Figures and Tables

**Figure 1 antibiotics-12-00935-f001:**
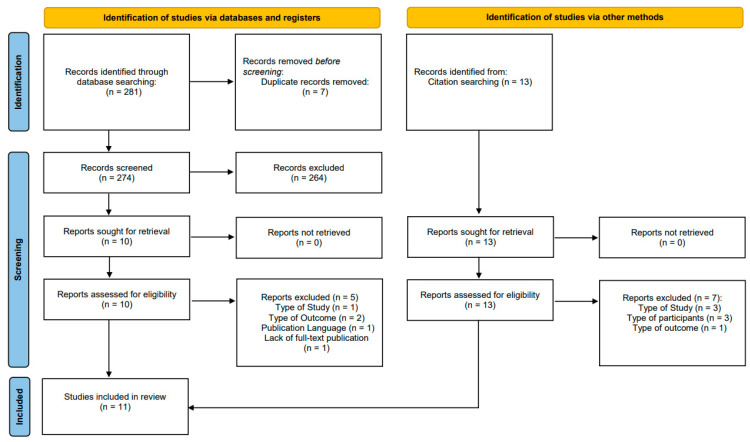
Flow diagram for identification of studies via databases, registers, and citation searching.

**Figure 2 antibiotics-12-00935-f002:**
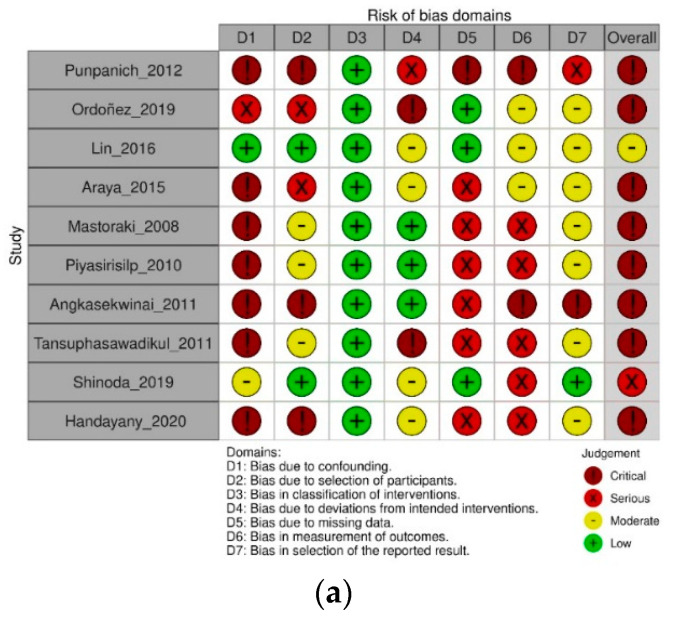
Detailed information on risk of bias as evaluated with (**a**) ROBINS -I and (**b**) RoB-2. Punpanich 2012 [23], Lin 2016 [24], Araya 2015 [22], Mastoraki 2008 [25], Piyasirisilp 2010 [27], Angkasekwinai 2011 [21], Tansuphasawadikul 2011 [28], Shinoda 2019 [29], Ordoñez 2019 [30], Hadayany 2020 [26], Snyman 2009 [6].

**Figure 3 antibiotics-12-00935-f003:**
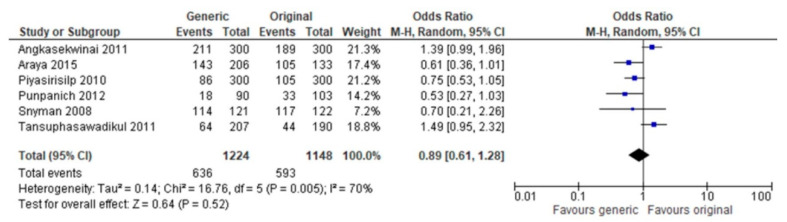
Analysis of clinical cure in studies comparing generic and branded antibiotics. Punpanich 2012 [23], Araya 2015 [22], Piyasirisilp 2010 [27], Angkasekwinai 2011 [21], Tansuphasawadikul 2011 [28], Snyman 2008 [6].

**Figure 4 antibiotics-12-00935-f004:**
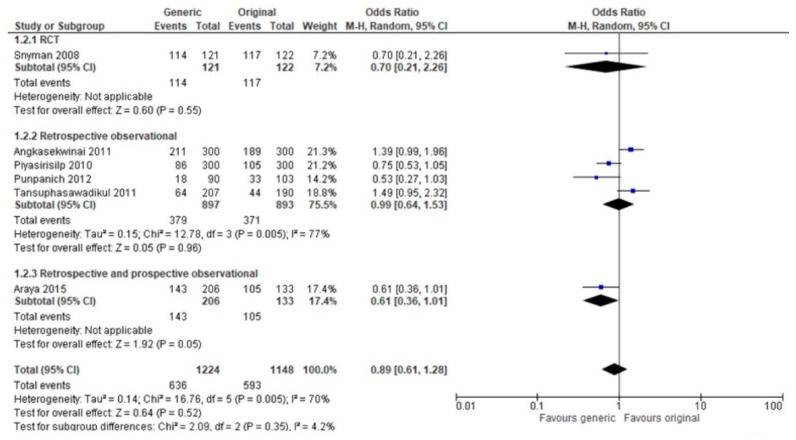
The effect of generic versus branded antimicrobial on clinical cure stratified by study design. Punpanich 2012 [23], Araya 2015 [22], Piyasirisilp 2010 [27], Angkasekwinai 2011 [21], Tansuphasawadikul 2011 [28], Snyman 2008 [6].

**Figure 5 antibiotics-12-00935-f005:**
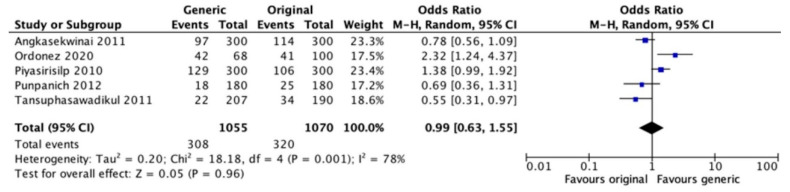
Analysis of overall mortality after treatment with generic or branded antimicrobial formulation of carbapenems. Punpanich 2012 [23], Ordoñez 2019 [30], Piyasirisilp 2010 [27], Angkasekwinai 2011 [21], Tansuphasawadikul 2011 [28].

**Figure 6 antibiotics-12-00935-f006:**
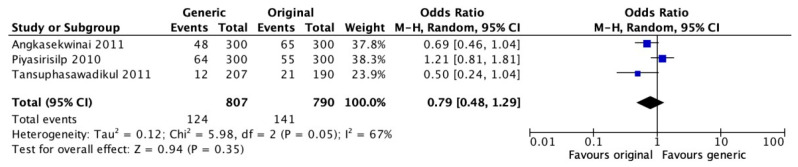
Analysis of death associated with infections after treatment with generic or branded antimicrobial formulation of carbapenems. Piyasirisilp 2010 [27], Angkasekwinai 2011 [21], Tansuphasawadikul 2011 [28].

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
