# Peer review of "Clinical Equivalence between Generic Versus Branded Antibiotics: Systematic Review and Meta-Analysis"

_antibiotics, 2023, doi:10.3390/antibiotics12050935_

Round 1

Reviewer 1 Report

I think that in order to compare a brand name antibiotic and its generic, one should collect data on their relationship with bacteria, on their resistance profiles. These are the parameters that will mainly influence the effectiveness of the clinical and microbiological treatment. For me the parameters chosen by the author are not relevant to justify the relevance of his study.

My comments

Line 80-91 : A doctor normally prescribes an antibiotic to treat a disease after an examination that identifies the bacteria involved and its reaction to a number of antibiotics (antibiogram). Normally we know the name of the bacteria with the antibiogram of this bacteria. What disturbs me here is that the author does not give these details. One does not test the clinical effectiveness of an antibiotic if one does not know the type of Gram to which the bacteria belong. Antibiotics are not given like that

Line 82-84 : I think that the mode of action, spectrum of activity and bacteriological effect should also be taken into account when comparing a generic antibiotic and its branded equivalent ;

Line 78-123 : What bothers me about the methodology used is that it is not clear whether the scientific work included in this review is clinical trials or simply work on antibiotic susceptibility testing where brand name or generic antibiotics were used.

Line 113-117 : Data on treated diseases and their bacterial causes (which bacteria) are missing. And the effectiveness of an antibiotic varies depending on the antibiotic in relation to the bacteria. Also the author does not say if the bacteria causing these diseases were sensitive to brand name antibiotics or generic ones. Because if an antibiotic is used for which the bacterium in question carries an intrinseque resistance, this antibiotic will not be effective.

Author Response

Dear reviewer, thank you for your contribution and questioning to the study. Your contributions were of great importance. We hope that the answers and our considerations are satisfactory.

I think that in order to compare a brand name antibiotic and its generic, one should collect data on their relationship with bacteria, on their resistance profiles. These are the parameters that will mainly influence the effectiveness of the clinical and microbiological treatment. For me the parameters chosen by the author are not relevant to justify the relevance of his study.

The aim of the study was to review the efficacy based on clinical cure or clinical meaningful outcomes in the literature. I agree that it is important to isolate the bacterial agents that cause the infection we are treating, in order to direct and optimize the treatment. Unfortunately, in many cases it is not possible to identify this agent, so the treatment ends up being empirical, based on the microbiological profile of each unit, hospital and city, for a given site of infection.

Few articles found evaluate microbiological cure, the main outcome being clinical cure and mortality. The patients being compared are being treated at the same site of infection, and in the same hospital, one group with generics and the other with original antibiotics, so despite the possibility of treatment failure due to bacterial resistance, this risk is possible in both groups.

Line 80-91 : A doctor normally prescribes an antibiotic to treat a disease after an examination that identifies the bacteria involved and its reaction to a number of antibiotics (antibiogram). Normally we know the name of the bacteria with the antibiogram of this bacteria. What disturbs me here is that the author does not give these details. One does not test the clinical effectiveness of an antibiotic if one does not know the type of Gram to which the bacteria belong. Antibiotics are not given like that

Dear reviewer, thank you for your comment, we agree on the importance of identifying the pathogen that causes the infection in therapeutic success, however the sensitivity of bacterial cultures is low and quite varied, depending on the site of infection, with treatment guided by the microbiological profile of each institution to a particular infectious site. The purpose of the article was precisely to analyze real-life scenarios, comparing the clinical equivalence of the treatment of infections between generic and original antibiotics. For approval of generic antibiotics, regulatory agencies require studies comparing pharmaceutical equivalence and bioequivalence of generic antibiotics with the original ones, and in healthy patients, with single doses. Therefore, we believe it is relevant to evaluate comparisons between these drugs in real-life studies, evaluating the treatment of infections, with outcomes of clinical cure and mortality.

Line 82-84 : I think that the mode of action, spectrum of activity and bacteriological effect should also be taken into account when comparing a generic antibiotic and its branded equivalent;

We consider that the spectrum of activity of approved generic antibiotics is similar to the original ones. The bacteriological effect was evaluated in vitro studies, and we consider it extremely important, however the objective of our review was to analyze the literature of studies comparing clinical efficacy in the treatment and prophylaxis of infections, and in this way to consider drug interactions, pharmacodynamics in patients with infectious diseases, in different profiles of severity and comorbidities.

Line 78-123 : What bothers me about the methodology used is that it is not clear whether the scientific work included in this review is clinical trials or simply work on antibiotic susceptibility testing where brand name or generic antibiotics were used.

About the methodology, in vitro studies or works that evaluated only susceptibility tests were not included, the objective was to include comparative studies between generic antibiotics and the corresponding branded drugs regarding their clinical equivalence.

In this study, clinical equivalence was defined by the similarity between treatments in comparative studies for human treatment or prophylaxis when considering clinical meaningful outcomes.

The research question that guided this systematic review was: What is the efficacy, safety and clinical equivalence of generic antibiotics in relation to corresponding branded drugs?

Due to the paucity of studies found in a validation search, any study design was considered for inclusion as long as it provided a comparison between generic and its respective branded formulation in a superiority or non-inferiority study. We excluded phase II randomized clinical trials, bioequivalence and pharmacokinetic trials, animal models and in vitro studies.

Line 113-117 : Data on treated diseases and their bacterial causes (which bacteria) are missing. And the effectiveness of an antibiotic varies depending on the antibiotic in relation to the bacteria. Also the author does not say if the bacteria causing these diseases were sensitive to brand name antibiotics or generic ones. Because if an antibiotic is used for which the bacterium in question carries an intrinseque resistance, this antibiotic will not be effective.

Thank you for your considerations, we agree with the importance of including information on identification and sensitivity to antimicrobials of the infections analyzed in the studies. We included microbiological information from the studies in the article, between lines 213-249. 

Three studies, Handayany et al. (2020), Lin et al. (2006), Shinoda et al. (2019), did not report microbiological data, none of which were used in the meta-analysis. The other studies reported microbiological data. Below are some microbiological data reported by the studies used in the meta-analyses.

Angkasekwinai et al. (2011) isolated microorganisms from two-thirds of the patients. The most common pathogen was gram-negative bacteria including extended spectrum- beta-lactamase (ESBL) producing E. coli and K. pneumoniae, Pseudomonas aeruginosa and Acinetobacter baumannii. Susceptibility to meropenem of these isolates was available in 96% percent. All isolates of ESBL producing E. coli and K. pneumoniae were susceptible to meropenem. No significant difference in susceptibility to meropenem for Pseudomonas aeruginosa was found between original and generic group. For Acinetobacter baumannii, the isolates in the original group were more susceptible to meropenem than those in the generic group (30% vs.8.1%, p = 0.02). 

Araya et al. (2015) excluded from the study sepsis caused by gram-positive bacteria, fungal infections. There was an approximately equal distribution between the groups in terms of percentage of gram negative.

Piyasirilp et al. (2010) observed microbiological documented infections in two-thirs of the patients. All isolates of extended-spectrum-beta-lactamase producing E. coli and K. pneumoniae were susceptible to imipenem/ cilastatin. P. aeruginosa isolates were susceptible to imipenem in 84.8% and 82.9% of the patients in the original imipenem/cilastatin group and the generic imipenem/cilastatin group, respectively. Acinetobacter baumannii isolates were susceptible to imipenem in 65.5% and 60% of the patients in the original imipenem/ cilastatin group and the generic imipenem/cilastatin group, respectively. The characteristics of infections and causative agents of infections of the patients in both groups were not significantly different.

Snyman et al. (2009) collected approximately 164 respiratory samples (110 throat swabs and 54 sputum specimens) from patients with tonsillitis and pneumonia. Organisms cultured at visit 1 included S. pyogenes, H. influenzae, S. aureus, S. pneumoniae and H. parainfluenzae. The results of the bacteriological cure rates for both the ITT and PP populations showed similarly high rates, with 39 (84.8%) in the original Clarithromycin arm and 31 (79.5%) in the generic Clarithromycin arm for the ITT population experiencing bacteriological cure. In the PP population, the bacteriological cure rate for original Clarithromycin was 37 (84.1%), while that for generic Clarithromycin was 31 (81.6%).

In Tansuphasawaikul et al. (2011) the three most common causative microorganisms were Pseudomonas aeruginosa, Acinetobacter baumannii, and ESBL producing Escherichia coli. There were no statistical differences of the distribution of the sites of infection and causative microorganisms between the two groups, except for ESBL-producing E. coli, which was significantly higher in the generic meropenem group.

Ordonez et al. (2019) included only patients with documented gram-negative infections susceptible to meropenem to maximize the likelihood of successful treatment with the selected antibiotics.

Punpanich et al. (2012) observed microbiologically confirmed infections in 40% and 32% among generic and original meropenem group, respectively. Including all sites of positive culture, the rates identification (infection/colonization) of extended spectrum beta-lactamase (ESBL) producing gram negative enteric bacteria and multi-drug resistant (MDR) Acinetobacter baumannii were not significantly different between the two treatment groups. ESBL producing bacteria were identifi ed in 25 cases (13.8%) and 24 cases (13.3%) among generic and original meropenem group, respectively (p = 0.878). In addition, MDR A. baumannii were identified in 11 cases (6.1%) and 15 cases (8.3%) among generic and original meropenem group, respectively (p = 0.415). Microbiological cure rates were also comparable: 70.4% and 70.8% (absolute difference: 0.4%, 95% CI: -25.5%, 24.58%) on day 3; and 80.9% and 81.3% (absolute difference: 0.4%, (95% CI: -25.7%, 25.15%) day 7 for the generic and original group, respectively.

Dear reviewer,  thank you for your contribution. We attach below the new version of the article with the suggested improvements, the changes were marked using the "Trach Changes" function of microsoft word.

Kind regards,

André Cotia

Reviewer 2 Report

The systematic review by Cotia et al provides a clinical perspective on the comparison of generic and branded antibiotics. Overall, the authors have done a good job of summarising the findings of previous studies. I firmly believe this review will be of utmost importance for researchers in the field of antibiotics and antimicrobial resistance. However, I do have one major comment and a few minor comments for the authors to address.

Major comment

I would suggest the authors to kindly review the manuscript for any typos or sentences that don't make sense, I have highlighted a few in the minor comments, but a careful read would reduce the chances of any mistakes that I couldn't detect.

Minor comments

In the abstract, italicize "in vitro"

Line 156, add "the" before "participants".

Line 157, add "to" after "due".

Line 158, add "the" before "selection".

Line 180-181, "fewer patients under immunosuppressive drugs when compared with its respective generic, both statistically significant", it doesn't make sense I think the sentence needs some rephrasing. 

Line 183,  what is the meaning of "Error! Bookmark not defined".

Line 250, replace "scarce" with "rare".

Line 264, add  "in which" before 

Line use "in vivo studies".

Lin 261 -262, rephrase the sentences to "Seven of the eleven studies were on β-lactam antibiotics, and each one compared glycopeptides, fluoroquinolones, trimethoprim-sulfamethoxazole, and macrolides"

I think English editing is not required.

Author Response

The systematic review by Cotia et al provides a clinical perspective on the comparison of generic and branded antibiotics. Overall, the authors have done a good job of summarising the findings of previous studies. I firmly believe this review will be of utmost importance for researchers in the field of antibiotics and antimicrobial resistance. However, I do have one major comment and a few minor comments for the authors to address.

Major comment

I would suggest the authors to kindly review the manuscript for any typos or sentences that don't make sense, I have highlighted a few in the minor comments, but a careful read would reduce the chances of any mistakes that I couldn't detect.

Dear reviewer, thank you for your comments and suggested text corrections. We apologize for the errors, we revised the article and corrected some sentences and phrases.

Dear reviewer, thank you for your comments, we apologize for the errors, we revised the article and corrected some sentences and phrases.

Minor comments

In the abstract, italicize "in vitro"

Line 156, add "the" before "participants". 

Line 157, add "to" after "due".

Line 158, add "the" before "selection".

Line 180-181, "fewer patients under immunosuppressive drugs when compared with its respective generic, both statistically significant", it doesn't make sense I think the sentence needs some rephrasing. 

Line 183,  what is the meaning of "Error! Bookmark not defined".

Line 250, replace "scarce" with "rare".

Line 264, add  "in which" before 

Line use "in vivo studies".

Lin 261 -262, rephrase the sentences to "Seven of the eleven studies were on β-lactam antibiotics, and each one compared glycopeptides, fluoroquinolones, trimethoprim-sulfamethoxazole, and macrolides"

We have corrected all the errors mentioned above by the reviewer.

We attach below the new version of the article with the suggested improvements, the changes were marked using the "Trach Changes" function of microsoft word.

Kind regards,

André Cotia

Round 2

Reviewer 1 Report

Some comments

Line 226 : You shoul write Gram-negative instead of gram-negative Please correct this also throughout the document.

Line 221-223 : This paragraph does not make sense in this document. It is well known that fungal infections are not caused by bacteria, although some mechanisms of resistance to antifungals and antibiotics may be similar. I think this paragraph should be removed from the document.

Author Response

Dear reviewer, thank you for the new comments and suggestions. Follow below the answers.

Line 226 : You shoul write Gram-negative instead of gram-negative Please correct this also throughout the document.

We have corrected all "Gram-negative" words in the article.

Line 221-223 : This paragraph does not make sense in this document. It is well known that fungal infections are not caused by bacteria, although some mechanisms of resistance to antifungals and antibiotics may be similar. I think this paragraph should be removed from the document.

We removed the phrase about the exclusion criteria of the patients with proven fungal infections in the study by Araya et. al. (2015).

We attach below the new version of the article with the suggested improvements.

Kind regards,

André Cotia
